# Foliar Calcium Absorption by Tomato Plants: Comparing the Effects of Calcium Sources and Adjuvant Usage

**DOI:** 10.3390/plants12142587

**Published:** 2023-07-08

**Authors:** Eduardo Santos, Gabriel Sgarbiero Montanha, Luís Fernando Agostinho, Samira Polezi, João Paulo Rodrigues Marques, Hudson Wallace Pereira de Carvalho

**Affiliations:** 1Group of Specialty Fertilizers and Plant Nutrition, Laboratory of Nuclear Instrumentation, Centre for Nuclear Energy in Agriculture, University of São Paulo, Avenida Centenário, 303, Piracicaba 13400-970, Brazil; eduardosr07@usp.br (E.S.); gabriel.montanha@usp.br (G.S.M.); joaoanatomia@gmail.com (J.P.R.M.); 2Laboratory of Functional Genomics and Proteomics of Model Systems, Department of Biology and Biotechnology, Sapienza University of Rome, Via dei Sardi, 70, 00185 Rome, Italy; 3Luiz de Queiroz College of Agriculture, University of São Paulo, Avenida Pádua Dias, 11, Piracicaba 13418-900, Brazil; lf.agostinho@usp.br (L.F.A.); sapolezi.silva@usp.br (S.P.); 4Department of Basic Science, Faculty of Animal Science and Food Engineering, University of São Paulo, Pirassununga 13635-900, Brazil

**Keywords:** calcium, foliar fertilization, X-ray fluorescence spectroscopy, blossom end rot, xylem, phloem

## Abstract

The deficiency of calcium (Ca) reduces the quality and shelf life of fruits. In this scenario, although foliar spraying of Ca^2+^ has been used, altogether with soil fertilization, as an alternative to prevent deficiencies, little is known regarding its absorption dynamics by plant leaves. Herein, in vivo microprobe X-ray fluorescence was employed aiming to monitor the foliar absorption of CaCl_2_, Ca-citrate complex, and Ca_3_(PO_4_)_2_ nanoparticles with and without using adjuvant. We also investigated whether Sr^2+^ can be employed as Ca^2+^ proxy in foliar absorption studies. Moreover, the impact of treatments on the cuticle structure was evaluated by scanning electron microscopy. For this study, 45-day-old tomato (*Solanum lycopersicum* L., cv. Micro-Tom) plants were used as a model species. After 100 h, the leaves absorbed 90, 18, and 4% of aqueous CaCl_2_, Ca-citrate, and Ca_3_(PO_4_)_2_ nanoparticles, respectively. The addition of adjuvant increased the absorption of Ca-citrate to 28%, decreased that of CaCl_2_ to 77%, and did not affect Ca_3_(PO_4_)_2_. CaCl_2_ displayed an exponential decay absorption profile with half-lives of 15 h and 5 h without and with adjuvant, respectively. Ca-citrate and Ca_3_(PO_4_)_2_ exhibited absorption profiles that were closer to a linear behavior. Sr^2+^ was a suitable Ca^2+^ tracer because of its similar absorption profiles. Furthermore, the use of adjuvant affected the epicuticular crystal structure. Our findings reveal that CaCl_2_ was the most efficient Ca^2+^ source. The effects caused by adjuvant suggest that CaCl_2_ and Ca-citrate were absorbed mostly through hydrophilic and lipophilic pathways.

## 1. Introduction

Calcium (Ca) is a macronutrient that plays both structural and physiological roles in vegetal metabolism [1]. In plant cells, Ca is found in its cationic form (Ca^2+^) and notably encompasses the cell wall, as well as abiotic and biotic stress signaling pathways [2].

Calcium ions are admitted through Ca^2+^-permeable channels in the plasma membrane, and its cytosolic concentration is tightly regulated via Ca^2+^-ATPase and anti-Ca^2+^ proteins [1,3] once Ca^2+^ variations may either trigger the plant’s immune defense system or lead to the formation of calcium phosphate [4,5]. Thereby, the free-Ca^2+^ concentration is significantly higher in the vacuoles than in the cytosol (*ca*. 1–10 mM and 100–200 nM, respectively) [6,7] whereas ca. 60% of the Ca content in plant cells is calcium pectate, i.e., a strong bond between Ca^2+^ and homogalacturonan, rhamnogalacturonan, and xylogalacturonan pectins anchored at the middle lamella of the cell wall [2,8].

In this context, some studies show that Ca^2+^ remobilization through the phloem is virtually non-existent [2,9]. Several studies have shown that, in contrast to other nutrients, Ca^2+^ concentration does not decline in leaves at the senescence stage and might often increase due to dehydration [10].

Hence, most of the Ca^2+^ supply to plants’ organs occurs solely through the xylem, which is driven by transpiration and is mainly dependent on soil Ca^2+^ availability [11]. From an agronomic standing point, this is a particular issue for providing optimal Ca^2+^ content in fruit species. Early developing fruits are served both by xylem and phloem flows, but growing fruits experience a gradual decrease in the number of functional xylematic vessels, as well as in transpiration rate, to the extent that the phloem becomes an exclusive source of nutrient supply leading to a dilution in Ca^2+^ concentration in the tissue [12].

Several physiological disorders, such as bitter pit, wilting, and apical rot, are frequently attributed to Ca^2+^ deficiency in important fruit species, such as apple (*Malus domestica*), watermelon (*Citrullus lanatus*), pepper (*Capsicum annuum*), and tomato (*Solanum lycopersicum*) [12,13]. Although the molecular mechanisms underlying these symptoms remain unclear, Ca^2+^ deficiency yields less rigid cell walls, which affect homeostasis and trigger the transduction of chemical signals that cause tissue necrosis [5,14], thus reducing the fruit’s postharvest quality and market value.

Hence, increasing Ca^2+^ phloem mobility may spawn foliar-based alternatives to meet plant requirements during fruit development. Phloem loading is a dynamic process that varies among plant tissues and species. In leaves, most of the photoassimilates present in the mesophyll cells move symplastically through the plasmodesmata toward the sieve-tube elements and companion cells surrounding the leaf veins [15,16,17].

This process is controlled by osmotic pressure differences in apoplast cells, according to the Münch model [18]. In principle, one may hypothesize that either high-soluble calcium sources or their combination with high phloem-mobility molecules might induce Ca^2+^ remobilization. Nevertheless, because Ca^2+^ is often defined as a non-mobile element, its dynamics during phloem loading have not been properly investigated [2,18].

Although the foliar application of Ca is a common practice used by fruit growers to prevent Ca-related disorders that affect fruit, Ca foliar absorption and transport have not yet been clarified. In this sense, the present study aimed at monitoring the foliar absorption of distinct Ca sources, that is, salt CaCl_2_ (Ca-chloride), nanosized Ca_3_(PO_4_)_2_ (Ca-phosphate), and Ca-citrate chelate, by tomato plants. We also investigated the effect of agriculture mineral oil adjuvant on the absorption process and the effect on the treatments on the leaf surface. Moreover, mineral oil plays the role of an adjuvant for fertilizer, fungicides, and herbicides [19,20]; adjuvant has been used as a pesticide in fruits and vegetables [21,22,23].

## 2. Results

### 2.1. Characterization of Ca:Sr Sources by ICP-OES

Powdered Ca:Sr- citrate and Ca:Sr- phosphate compounds were analyzed by ICP OES to determine the proportion of Ca:Sr and its major elements (Appendix A). Additionally, Ca:Sr- chloride solution was also analyzed as a reference compound. The proportion of Ca:Sr was calculated using the recorded Ca and Sr concentration (mol kg^−1^), which revealed that a slightly different proportion of Ca:Sr was present in the compounds (Table 1). In this regard, the solutions or dispersions were normalized to a 0.1 M Ca concentration before their use in the foliar absorption experiments. The recovery of certificate material is presented in the Appendix A.

### 2.2. Characterization of Ca:Sr- Citrate by FTIR

Fourier transform infrared spectroscopy (FTIR) was used to assess the Ca:Sr- citrate synthesized. The FTIR spectra presented in Figure 1 show changes in the absorption bands of the carboxyl group region (1600–1800 cm^−1^) for the Ca:Sr- citrate, where the absorption band was displaced to minor energy. As denoted by the gray and blue stripes, the absorption band at 1728 cm^−1^ encompasses a characteristic stretching vibration of C=O. Since the same band is observed at 1593 cm^−1^ in the citric acid spectrum, it is possible to assume that the reduction in absorption energy observed in the Ca:Sr- citrate spectrum indicates the interaction between the carboxyl group and Ca^2+^ [24,25].

### 2.3. Monitoring the Foliar Absorption of Ca:Sr Sources by µ-XRF

Microprobe X-ray fluorescence spectroscopy (µ-XRF) was employed to determine the foliar absorption of the Ca:Sr from citrate, chloride, and phosphate sources with or without adjuvant addition. The Ca:Sr- chloride exhibited the highest absorption at 100 h after the exposure, followed by citrate, and phosphate sources (Figure 2). Curiously, the use of adjuvant reduced the total absorption of chloride sources and increased that of citrate sources, which showed intraspecific statistical significance at 95%. On the other hand, it did not affect the total absorption in plants exposed to Ca:Sr- phosphate.

Figure 3 presents the in vivo absorption of Ca and Sr as a function of time. The total absorption rates were calculated for the three Ca:Sr sources, i.e., higher for the Ca:Sr- chloride than for the citrate and phosphate sources. It also shows that adjuvant usage affected the absorption behavior for Ca:Sr- chloride source, where the addition of mineral oil increased the absorption velocity during the initial 24 h after the exposure, followed by the stabilization phase. However, one should notice that despite exhibiting a slower absorption rate, a higher total absorption was observed after 96 h without adjuvant usage (Figure 3a,b). The absorption profile of the Ca:Sr- citrate source was also increased by adjuvant usage (Figure 3c,d). On the other hand, Ca:Sr- phosphate absorption was not affected by the addition of the adjuvant (Figure 3e,f).

The Ca:Sr absorption rates resulting from chloride and citrate sources can be explained through an exponential decay mathematical function (Equation (S1)). Nevertheless, since only the fitting of the Ca:Sr- chloride sources either with or without adjuvant usage presented correlation values (r^2^) above 80%, the values found for the other sources will not be herein discussed.

As shown in Table 2, the adjuvant usage increased the absorption rate (Abs. rate (Equation (S4))) and reduced the absorption half-life, i.e., the time required for the plant to absorb 50% of fertilizer applied (T_50%_) (Equation (S2)), and the fraction of fertilizer absorbed at the moment in which absorption kinetic is reduced (Abs. F, calculated by subtracting y0 of A1 (Equation (S3))) in the chloride source**.**

Figure 4 shows µ-XRF chemical maps of the Sr spatial distribution in tomato leaves exposed to Sr-citrate formulation. One can note an agglomeration of Sr citrate that caused an increase in the Sr intensity at 26 h after the application. This clustering explains the increase in the amount of Ca and Sr observed during the 100 h absorption experiment (Figure 3). Additionally, it is important to highlight that the changes Ca intensity in the leaf tissues as a function of Sr exposure is due to Sr-induced auto-absorption of incoming and outgoing X-ray photons, a characteristic phenomenon of the XRF technique.

Furthermore, Figure 5 shows µ-XRF chemical maps of the Sr spatial distribution of tomato leaves exposed to Sr-chloride formulation. It showed a gradual reduction of Sr intensity in tomato leaves, thereby confirming its absorption. Although these results do not enable assessing whether Sr was transported to another part of the plants, it reinforces that the chloride exhibited a higher foliar absorption than the citrate. For comparison purposes, chemical map results were summarized in Table 3.

### 2.4. Ultrastructural Characterization of Sr:Ca Sources on Tomato Leaves by SEM

Figure 6 shows the ultrastructural characteristics of tomato leaflet cuticles exposed to the different Ca:Sr- sources, as well as the 1% adjuvant herein explored. We observed that the control tomato leaf surface is composed of epidermal lens-shaped cells with striated wax cuticles (Figure 6a–c). The use of the adjuvant diluted in water demonstrates that this solution is able to cover both anticlinal and periclinal epidermal cell walls uniformly. However, the adjuvant caused the loss of the striated cuticle wax patterns (Figure 6d–f).

SEM analysis revealed small clusters of residues from the applied formulation over the leaf surface (Figure 6g–o). Among all Ca sources, only Ca:Sr- chloride was able to reduce the leaf striate pattern in regions at the cuticle wax, leading to a smooth surface on some regions of the periclinal cell wall of the epidermal cell (Figure 6h,i). Calcium citrate was observed as platelets crystals (Figure 6j–l) e Ca phosphate presented powder crystals covering the surface, but no drastic changes of the striated patterns were observed (Figure 6n,o).

## 3. Discussion

In this study, the effects of adjuvant usage on the foliar absorption dynamics of three Ca sources were evaluated by in vivo XRF. This novel strategy was carried out using Sr^2+^ as a physiological tracer of Ca^2+^ in plant tissues. Plants cannot strongly distinguish between both ions due to their similar chemical properties, such as oxidation state and ionic radii [26], and exhibit virtually no background signal in plant materials. Evidence of Sr partially replacing Ca in plant metabolism has been reported since 1937 [27,28,29,30] and used in several species, e.g., tomato [31], citrus [32], apple [33], and corn [34]. Our results revealed very similar absorption profiles of both Ca and Sr (Figure 3, Table 2), regardless of the source, thereby proving its feasibility.

Nevertheless, one should keep in mind that this approach does not allow the assessment of whether the applied fertilizer was transported to other plant parts. Translocation studies would require the proper following of the fertilizer’s passage through the cuticle and epidermis and its subsequent loading onto the phloem toward other tissues [35], which would require complementary methods such as measuring the concentration or activity of stable or radioactive isotopes in different plant parts. Gomes et al. (2019) also employed in vivo XRF approach for monitoring the Zn concentration in the petiole of soybean plants [36].

Despite the lack of studies that measure the foliar absorption rate of Ca as a function of time, several studies indicate that foliar Ca application is an effective practice for preventing physiological disorders caused by Ca deficiency in fruit species [37,38,39,40,41], although the Ca transport in phloem is not fully understood. In this regard, Song et al. (2018) evidenced the phloem Ca transport [42,43]. Additionally, Bonomelli et al. (2019) conducted research on the effectiveness of foliar fertilization in providing calcium to fruits using 45Ca. The study found that only a small percentage, approximately 1%, of the applied calcium was transferred to the fruit. The majority of the Ca remained in the treated tissue [44]. Therefore, understanding the absorption process is crucial for Ca retention in the treated tissues. In the present study, Ca and Sr absorption followed the same order as the solubility of the treatments, i.e., chloride > citrate > phosphate [26,45,46]. Therefore, soluble sources are more efficiently absorbed [47]. This finding shows that tomato leaves do not take up all particles, such as nanoparticles [48]. Hence, the effectiveness of concentrated suspensions of low-Ksp compounds such as calcium phosphates must be carefully considered.

Furthermore, other factors resulting from the interaction between foliar fertilizer and leaf surface can influence absorption, such as wettability, adhesion, and surface tension [35]. Although we did not measure the superficial tension of the foliar fertilizers, it can be deduced that the adjuvant used changed the chemical properties of the foliar formulations [21,22,23], causing a reduction in surface tension and increasing adhesion on the leaf surface, thus changing the absorption kinetic profile.

Scanning electron microscopy (SEM) showed that only Ca:Sr- chloride reduced the striated pattern of the cuticle at the periclinal cell wall. Alterations in composition and deposition in the leaf cuticle can enhance the diffusion of Ca^2+^ in the cuticular matrix and cell walls because it is a hydrophobic protective layer that covers the epidermis. The striated pattern is composed of epicuticular wax on the surface that seems to be modified after the deposition of the calcium chloride droplet, whereas it was not observed in leaves exposed to both Ca:Sr- citrate and phosphate [49].

Similar conditions were observed by Machado et al., (2019) and Gomes et al., (2020), where soybean plants exposed either to soluble manganese or zinc sources also Ca exhibited dissolution of the epicuticular wax crystals, whereas an opposite pattern was found for low-absorbed sources, such as MnCO_3_ [50,51]. Therefore, it is possible to assume that the Ca:Sr- chloride source presented a higher interaction with the leaf surface cuticle and ultrastructural changes in the striated pattern of the cuticle at the periclinal cell wall. Additionally, transmission electron microscopy must be conducted to verify changes in the epidermal cell wall as well as the possible roles of leaf stomata in Ca absorption.

Herein, the absorption kinetics profile of Ca:Sr- chloride was explained by an exponential decay equation (Equation (S1)), whereas it could not explain those observed for other sources, thus indicating clear differences among them. Interestingly, Ca:Sr- chloride fitting revealed that adjuvant usage increased the absorption rate (Abs. Rate) by approximately nine-fold, and t_50%_ was reduced by approximately three-fold, which explains the higher total Ca and Sr absorption under adjuvant restriction [52]. A similar kinetic profile was observed in soybean leaves exposed to different K sources, where foliar absorption was positively correlated with the solubility of the compound [53].

To the best of our knowledge, this is the first study to report in vivo measurements able to determine the foliar absorption rate and the time required for the plant to absorb 50% of the applied fertilizer. Other strategies include assessing transport by measuring the activity of radioisotopes [44,54] or the concentration [38,39,55] of elements applied to different parts of the plant. Although these methods are highly accurate, they are unfeasible for in vivo measurements because of the laborious methodology, which requires the removal and detachment of tissue.

Nevertheless, foliar absorption is a complex process that varies as a function of the leaf tissue composition and environmental conditions. In this context, it is important to consider certain factors for establishing an adequate in vivo XRF-based study: (i) High air relative humidity conditions can induce fertilizer losses through the formation of droplets on the leaf surface. (ii) Excessive exposure to X-rays can cause damage to leaf tissues [56], which requires prior optimization of the instrumental parameters to avoid artifacts.

## 4. Material and Methods

### 4.1. Synthesis of Ca:Sr Sources and Preparation of Foliar Solutions

Given the high Ca^2+^ background in the shoot plants ranging from 1 to 50 g kg^−1^ dw, strontium (Sr^2+^) was herein employed as a physiological tracer for Ca^2+^, once both elements present akin physicochemical properties and are similarly assimilated by plants [31,57]. In this regard, three Sr-doped Ca sources were investigated: calcium citrate (Ca:Sr- citrate), calcium chloride (Ca:Sr- chloride), and calcium phosphate (Ca:Sr -phosphate) nanoparticles.

The Ca:Sr citrate was synthesized following the method proposed by Li et al. (2016) with some modifications [25]. Briefly, dispersions of Ca(OH)_2_ and Sr(OH)_2_ at 0.14 mol L^−1^ [Ca] and 0.06 mol L^−1^ [Sr] were mixed at a 7:3 proportion with a 0.15 mol L^−1^ citric acid until reaching a pH 4.7. Then, 50% *v*/*v* ethanol solution was added until the formation of the white slurry. The solution was kept standing for 12 h, then the white slurry was washed with deionized water and centrifuged at 7000 rpm three times, frozen in a refrigerator at −20 °C for 24 h, and finally, freeze-dried (Thermo Micro Modulyo 115 Freeze Dryer System) to obtain powder Ca:Sr- citrate.

The Ca:Sr- chloride was prepared by mixing CaCl_2_.2H_2_O and SrCl_2_.6H_2_O solutions at 0.1 mol L^−1^ [Ca] and 0.038 mol L^−1^ [Sr]. The nanosized Sr-doped calcium phosphate (hereby labeled as Ca:Sr- phosphate) was prepared according to the methodology described elsewhere Ramírez-Rodríguez, (2020) [58], and exhibited 24.38 ± 0.19%wt [Ca], and 11.38 ± 0.20%wt [Sr].

Lastly, Ca:Sr- citrate, Ca:Sr- chloride, and Ca:Sr- phosphate foliar solutions, or dispersions, were prepared at 0.1 mol L^−1^ [Ca]. Agricultural mineral oil at 1% (*v*/*v*) was added to the formulation as an adjuvant in order to investigate its effects on foliar absorption. Appendix A details the composition of all Ca:Sr formulations with and without adjuvant herein prepared.

### 4.2. Characterization of Ca:Sr Sources

The concentration of Ca and Sr in Ca:Sr- citrate and Ca:Sr- phosphate were determined according to the method proposed by RODRIGUES et al., (2020) [59]. Briefly, 0.1 g of each Ca:Sr compound was transferred to Teflon tubes, where 2.5 mL sub-boiling 20% (*v*/*v*) HNO_3_ (Merck, USA) and 1.5 mL of 30% (m/m) of H_2_O_2_ (Dinâmica, SP-Brazil) were added. The materials were digested using a microwave-assisted system (Provecto Analítica, model DGT 100-Plus, Brazil) operating at 400 W for 3 min), 850 W for 30 min, and 320 W for 3 min, respectively. Then, the tubes were left to cool down, and the resulting solutions were transferred to centrifuges tubes, where the volume was adjusted to 20 mL using ultrapure water, and analyzed by the ICP OES system (Thermo Scientific, Waltham, MA, USA). The accuracy of the digestion procedures, as well as the ICP OES measurements, was determined by measuring the concentration of the probed elements in the blank solutions and certified reference material (CRM-Agro #c1005—sugarcane leaves and SRM-1515—apple leaves).

Furthermore, the Ca:Sr- citrate complexation was evaluated by Fourier-transform infrared spectroscopy (FTIR). In this regard, 1 mg of powdered sample was mixed with 300 mg of KBr and put in the glass desiccator for 12 days, with the replacement of silica gel every two days. Then, the mix (sample + KBr) was kept in an oven set at 50 °C for 24 h. After this time, the material was pressed, and the pellet was immediately analyzed by an FTIR spectrometer in transmission mode.

### 4.3. Plant Cultivation

*Solanum lycopersicum* cv. Micro-Tom plants were grown in a greenhouse for up to one week before analysis to acclimate them to the growth chamber. Tomato seeds were sown in 250 mL plastic pots containing a 1:1 soil-vermiculite mixture substrate amended with limestone and 4:14:8 NPK solution at 8 g L^−1^. The plants were irrigated by keeping the plastic trays continuously immersed in a trickle of water.

### 4.4. In Vivo XRF Evaluation Foliar Absorption

The absorption kinetics were determined using in vivo X-ray fluorescence spectroscopy (XRF) [36,53]. In this regard, 40-day-old tomato plants were transferred to acrylic sample holders particularly developed for this analysis (Appendix A), and 0.5 µL droplets of Ca:Sr- citrate, Ca:Sr- chloride, or Ca:Sr- phosphate solutions at 0.1 mol L^−1^ [Ca] with and without adjuvant added were deposited onto their leaves (Appendix A). The experiment was carried out with five independent biological replications, whereby a fully expanded leaf of each plant was treated on the adaxial surface, avoiding pipetting the drop onto the veins. After 20 min, once the droplet was dried, the samples were loaded into the XRF system (Orbis PC, Edax, USA), and the region containing the droplet was scanned at 2, 8, 16, 25, 37, 52, and 100 h after application using 16-point lines produced by a 1 mm collimated X-ray beam operating at 700 µA and 45 kV. At each point, the spectra were recorded for 5 s by a silicon drift detector with a dead time < 10%. In the interval of each measurement, the plants were maintained in a growth chamber with 12 h of photoperiod; at 25 °C, and the relative humidity was increased during the night period from 45 ± 10 to 75 ± 10.

The obtained Ca and Sr intensities were integrated and normalized to the values recorded in the first measurement to assess their absorption as a function of time (Appendix A), according to the methodology described by Corrêa et al. (2021) [53] and fitted as a function of a non-linear exponential decay equation described in Equation (S1).

Furthermore, the two sources exhibiting the best absorption performance were selected to assess their absorption profile without Ca complexation. For that, 0.1 mol L^−1^ Sr-citrate and Sr-chloride solutions containing 1% (*v*/*v*) adjuvant were prepared and applied to tomato leaves as previously described. After the droplet drying, the leaf tissues were in vivo mapped through XRF using a 32 × 25 pixel matrix by a 30 µm X-ray beam operating at 900 µA and 45 kV. The spectra were recorded for 2 s using a silicon drift detector with a dead time < 10%, and only the recorded values above the instrumental limit of detection, calculated according to the methodology described by Rodrigues et al. (2018) [60], were considered. All data analyses were performed in Origin software (version 2022, OriginLab, USA).

### 4.5. Scanning Electron Mycroscropy

The effect of foliar solutions of Ca:Sr- citrate, Ca:Sr- chloride, and Ca:Sr- phosphate with 1% (*v*/*v*) adjuvant and control water with at 1% (*v*/*v*) on the leaf surface was accessed by Scanning Electron Microscope (SEM). In this respect, treated leaves were detached 24 h after application, and a region of ca. 0.5 cm^2^ in which the drop was pipetted was excised using a razor blade, then metalized with gold, as described by Rodrigues et al., (2020) [59]. For instance, after 24 h of application of the drops, the samples were immersed in Karnovsky’s fixative solution and placed in a vacuum in 3 repetitions of 15 min, then they were dehydrated in an ethylic sequence in repetitions of 15 min in a vacuum of 1 × 30%; 1 × 50%; 1 × 70%; 3 × 90% and 4 × 100%. Afterward, the samples were taken to the critical point dryer (LEICA CPD 300, Wetzlar, Germany), sputter coated with gold (BAL-TEC SCD 050, New York, NY, USA), and finally examined with a scanning electron microscope (JEOL JSM IT 300, Tokyo, Japan) at 20 kV and digitally recorded at a working distance of 10 mm.

### 4.6. Statistical Analyses

All quantitative data drawn from a normally distributed population determined using the Shapiro–Wilk test at 0.05 level were subjected to a two-way analysis of variance (two-way ANOVA) followed by Tukey’s multiple comparison test at a 95% confidence interval (*p* < 0.05). The data analyses were performed using Origin (2022b version, OriginLab, USA) and Prism (version 9.4.0, GraphPad, USA)

## 5. Conclusions

The results presented herein show that the absorption efficiency of the Ca:Sr sources was mainly dependent on their solubility. In addition, adjuvant usage increased the absorption rate of chloride- and citrate-based sources, thereby proving that the proposed in vivo XRF-based strategy is suitable for tracking foliar absorption of calcium sources by tomato leaves. These results might shed light on the dynamics of foliar absorption by living plants and foster studies aimed at understanding the long-distance transportation process and its chemical environment.

## Figures and Tables

**Figure 1 plants-12-02587-f001:**
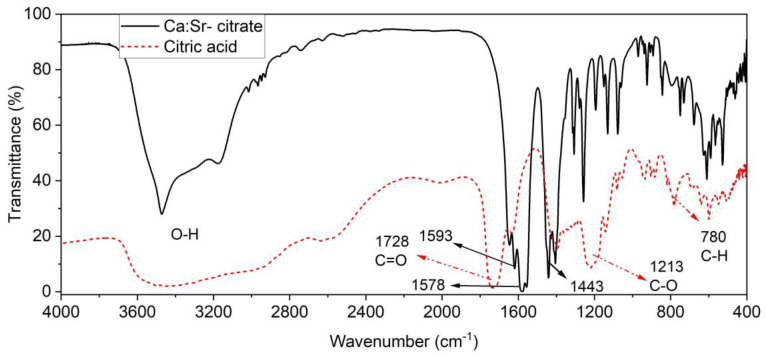
Shows Fourier transforms infrared spectroscopy (FTIR) of citric acid (Red dot line) and Ca:Sr- citrate (black line).

**Figure 2 plants-12-02587-f002:**
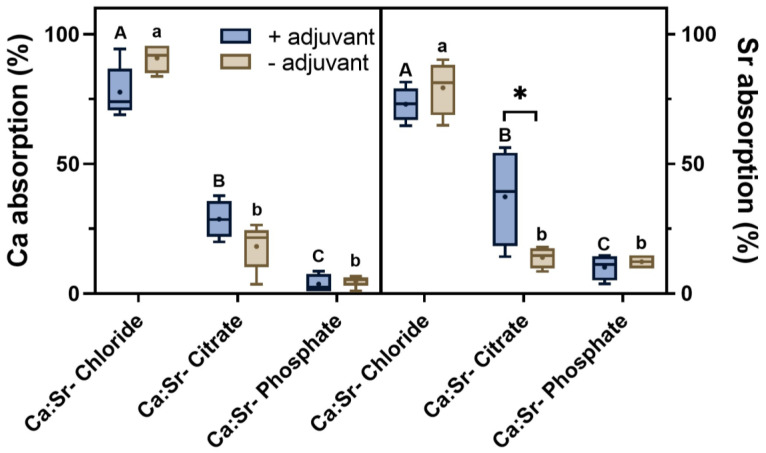
Calcium and Sr absorption by tomato leaves during a 100 h of exposure to 0.1 M Ca:Sr- chloride, citrate, or phosphate sources with or without 1% adjuvant addition. Data indicate the mean and the standard error from five independent biological replicates subjected to a two-way analysis of variance followed by Tukey’s test at a 0.05 significance level. Capital letters represent the comparison of the mean between sources with adjuvant. Lowercase letters represent the comparison of the mean between sources without adjuvant usage, asterisks denote the comparison of the mean between sources with and without adjuvant usage.

**Figure 3 plants-12-02587-f003:**
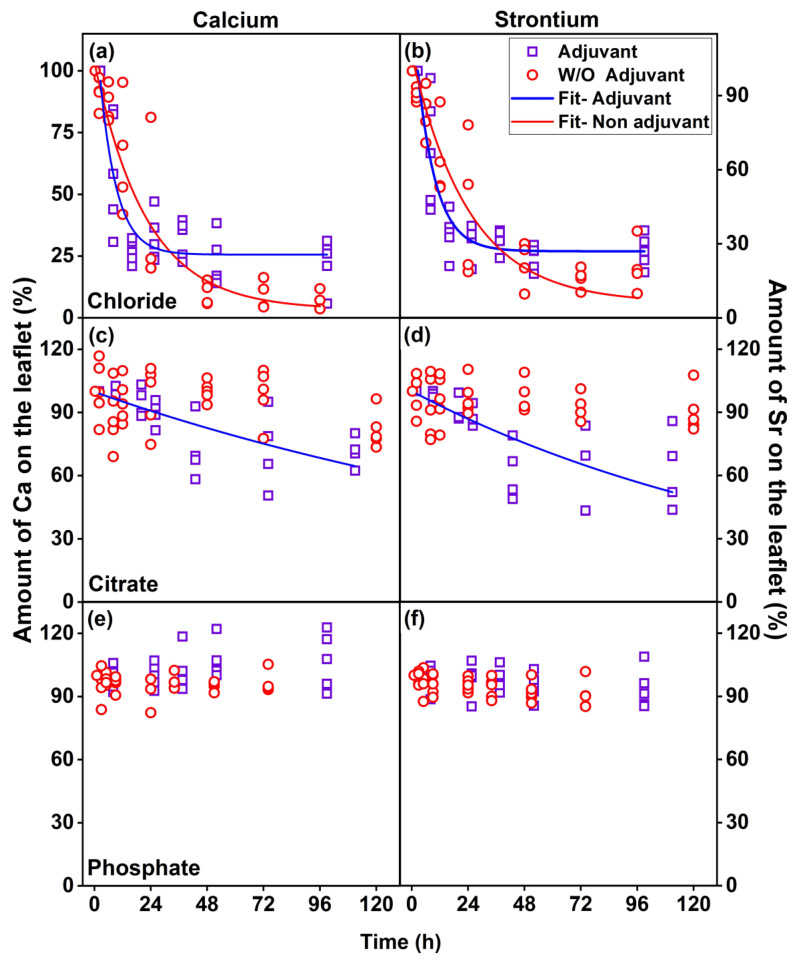
In vivo monitoring of the absorption kinetics of Ca (**a**,**c**,**e**) and Sr (**b**,**d**,**f**) by the leaves of tomato plants exposed to Ca:Sr- chloride (**a**,**b**), citrate (**c**,**d**), and phosphate (**e**,**f**) sources with or without adjuvant usage. The amount of Ca and Sr on the leaf was determined through integration of leaf areas evaluated through microprobe XRF. Data indicate the values above the instrumental limit of detection recorded on five independent biological replicates.

**Figure 4 plants-12-02587-f004:**
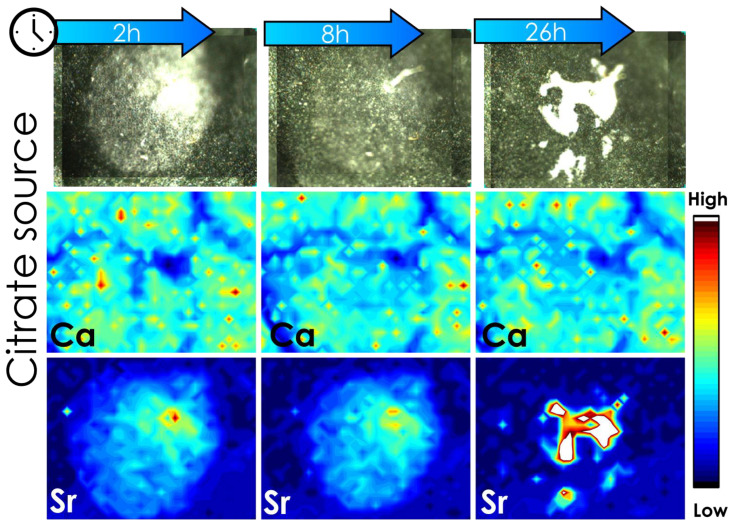
Photography of mapped area and its respective chemical map of Ca and Sr pass 2, 8, and 26 h of application. Here, 0.5 μL of Sr citrate fertilizer was dropped on the leaf and three XRF analyses were measured in the same region. The experiment was conducted with three biological replicates that showed a similar profile; figures of the other two replicates are shown in the Appendix A of the Appendix A. The intensity of Ca and Sr were adjusted to the same scale considering the intensity of the maps of 2 h after application.

**Figure 5 plants-12-02587-f005:**
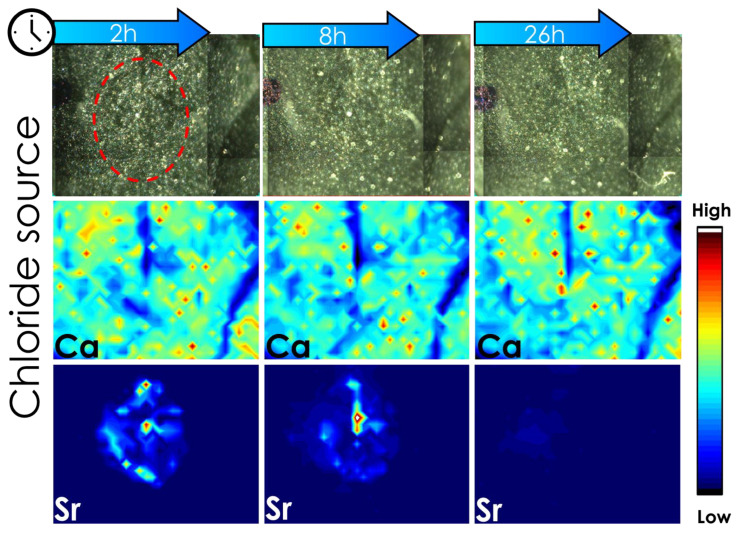
Photography of mapped area and its respective chemical map of Ca and Sr pass 2, 8, and 26 h of application, red circle highlights the drop. Here, 0.5 μL of Sr chloride fertilizer was dropped on the leaf and three XRF analyses were measured in the same region. The experiment was conducted with three biological replicates that showed a similar profile; figures of the other two replicates are shown in Appendix A of the Appendix A. The intensity of Ca and Sr were adjusted to the same scale considering the intensity of the maps of 2 h after application.

**Figure 6 plants-12-02587-f006:**
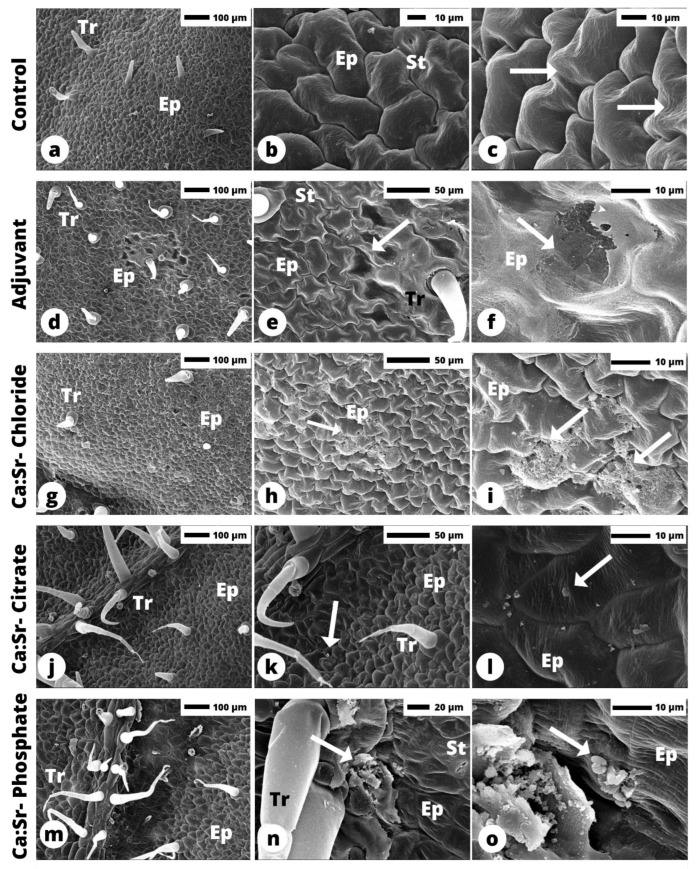
Scanning electron microscopy of tomato leaflets after the Ca sources were removed during the sample. See Control in (**a**–**c**); Adjuvant in (**d**–**f**); Ca:Sr- Chloride in (**g**–**i**); Ca:Sr- Citrate (**j**–**l**); Ca:Sr- Phosphate in (**m**–**o**). On the control (water) note the striated cuticle wax (arrows in (**c**)). The loose of the striated cuticle wax patterns was noticed in the Adjuvant treatment (arrows in (**e**,**f**)). The arrows in (**g**–**o**) represent the product residues. Ep - Epidermis; Tr - Trichome; St - stomata.

**Table 1 plants-12-02587-t001:** The concentration of Ca and Sr, in mol kg^−1^, recorded in Ca:Sr- citrate, Ca:Sr- chloride, and Ca:Sr- phosphate compounds.

Compound	Ca	Sr	Ca:Sr
	(mol kg^−1^)		
Ca:Sr- citrate	3.4 ± 0.03	0.540 ± 0.2	6.23
Ca:Sr- chloride	0.1 ± 0.001	0.014 ± 0.0	7.29
Ca:Sr- phosphate	0.4 ± 0.197	0.073 ± 0.3	5.46

**Table 2 plants-12-02587-t002:** Fitting results in absorption of calcium and strontium from the chloride source which Y0 refers to as horizontal asymptote, Abs rate- absorption rate, t50%- spent time to reduce 50% of what was applied, and Abs. F- the fraction of fertilizer absorbed at the moment in which absorption is kinetic. The equations to calculate these parameters are shown in Appendix A.

Calcium						
**Treatment**	**y0 (%)**	**A1 (%)**	**R^2^**	**Abs.** **Rate (h^−1^)**	**t_50%_** **(h)**	**Abs.F (%)**
Chloride + adjuvant	25.6 ± 2.9	100	0.8	0.14	4.9	74.4
Chloride − adjuvant	3.3 ± 4.6	100	0.9	0.045	15.4	96.7
Strontium						
Chloride + adjuvant	27.0 ± 2.6	100	0.9	0.13	5.5	73
Chloride − adjuvant	6.1 ± 4.3	100	0.9	0.041	17	93.9

**Table 3 plants-12-02587-t003:** The average amount of Sr on the tomato leaflet was obtained from three biological replicates. The Sr analytical signal of each chemical map was summed and normalized, considering the first measurement (2 h after application) as 100% providing the percentage of Sr on tomato leaves obtained from chemical maps 2, 6, and 26 h after foliar treatment without Ca. Data indicate the mean and the standard error from three independent biological replicates subjected to a one-way analysis of variance followed by Tukey’s test at a 0.05 significance level. Asterisks denote statistical difference between time of analysis.

Time (Hour)	Sr Chloride + Adjuvant (%)	Sr Citrate + Adjuvant (%)
2	100	100
6	92.1 ± 2.3	83.4 ± 7.8
26	* 37.8 ± 7.2	77.7 ± 7.6

## Data Availability

The raw data herein presented is fully available at the Figshare repository: https://doi.org/10.6084/m9.figshare.23638842.v1.

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
