# Peer review of "Foliar Calcium Absorption by Tomato Plants: Comparing the Effects of Calcium Sources and Adjuvant Usage"

_plants, 2023, doi:10.3390/plants12142587_

Round 1

Reviewer 1 Report

The paper follows a vewry interesting issue that might be interesting to the readers working with this scientific issue. The abstract has a clear aims and a general conclusion being well organized. The introduction furnishes a nice scientific background providing the most relevant points of the state of the art. Materials and methods allow reproduction of the experiments. Results seems robust with the discussion suporting these data. Conclusion although synthetic gives a simple picture of the paper.

Author Response

We would like to thank the Reviewer for taking the necessary time and effort to review the manuscript. We sincerely appreciate all your valuable comments.

Reviewer 2 Report

Dear Authors,

The subject of the study is interesting and topical, with scientific and practical importance.

The introduction is presented correctly, in accordance with the subject. Numerous scientific articles, in concordance to the topic of the study, were consulted.

Methodology of the study was clearly presented, and appropriate to the proposed objectives.

The obtained results are important and have been analyzed and interpreted correctly, in accordance with the current methodology. It is recommended that some issues be reviewed.

The discussions are appropriate, in the context of the results, and was conducted compared to other studies in the field.

The scientific literature, to which the reporting was made, is recent and representative in the field.

Some suggestions and corrections were made in the article.

The following aspects are brought to the attention of the authors.

1.

Please check the way of citing bibliographic sources recommended in Instructions for Authors, and Microsoft Word template, Plants journal.

e.g.

“[1]” instead of “1

It is recommended to check the entire content of the article and make the necessary corrections, if necessary.

2.

The structure of the article by chapters

According to Instructions for Authors, and Microsoft Word template, Plants Journal, the chapter structure is:

1. Introductions

2. Results

3. Discussions

4. Materials and Methods

5. Conclusions

Please check and organize the structure of the article.

It is possible that, through reorganization, the order of citing bibliographic sources in the content of the article will be changed.

This will require revising the References chapter to be consistent with the order of citing sources in the text.

3.

Consistency in writing

e.g.

Page 11, Figure 6

In the content of Figure 6, "a, b, c, d, e, f, g, h, i, j, k, l, m, n, o" are used.

In the title of Figure 6, "A, B, C, D, E, F, G, H, I, J, K, L, M, N, O" are used.

Also, in the text, in the content of the article, page 10, lines 299 to 305, "A, B, ..." are used.

Please check, to be consistent in writing.

4.

Writing some substances

e.g.

Page 11, title of the Figure 6

“CaCl2” instead of “CaCl2”

“Ca3(PO4)2” instead of “Ca3(PO4)2”

Page 13, rows 409, 411

“Ca2+” instead of “Ca2+”

5.

References

According to Instructions for Authors, and Microsoft Word template, Plants journal

Author 1, A.B.; Author 2, C.D. Title of the article. Abbreviated Journal Name Year, Volume, page range.

Include the digital object identifier (DOI) for all references where available.

e.g.

”White, P.J.; Broadley, M.R. Calcium in plants. Ann. Bot. 2003, 92, 487–511.”

Instead of

”White, P. J. & Broadley, M. R. Calcium in plants. Ann. Bot. 92, 487–511 (2003).”

It is recommended to check the entire References chapter, and correct where necessary.

Author Response

We would like to thank the Reviewer for taking the necessary time and effort to review the manuscript. We sincerely appreciate all your valuable comments and suggestions, which helped us in improving the quality of the manuscript.

Point-by-point comments

1- Please check the way of citing bibliographic sources recommended in Instructions for Authors, and Microsoft Word template, Plants journal.

Action: The way of citing the bibliographic references in the text was standardized to the rules of the Plants journal in the entire content of the article.

2- The structure of the article by chapters

Action: The structure of the article by chapters was changed considering the structure required for the Plants Journal. In this case, follow the current structure of the article. 1. Introductions; 2. Results; 3. Discussions; 4. Materials and Methods; 5. Conclusions.

3- Consistency in writing

Action: We changed all upper cases to lower cases in the title of Figure 6 and in the text. Also, we correct the scale of Figure 6. The alterations are highlighted in yellow in the revised article. 

4- Writing some substances

Action: The writing of the chemical form of the substances was revised and adjusted. The alterations are highlighted in yellow in the revised article. 

5- References

Action: The way of references was presented was revised and standardized to the rules of the Plants journal.

Reviewer 3 Report

The objective of this work should be more explicitly indicated in the abstract.

Likewise, in the introduction, the objectives of this research have not been clearly stated, in my opinion, so they should be added

The material and methods section seems correct to me.

Manuscript ID: plants-2467668 – Review

Manuscript ID: plants-2467668
Type of manuscript: Article
Title: Foliar calcium absorption by tomato plants: comparing the effects of
calcium sources and adjuvant usage.

  In the results section in Table 1, no letters , nor * have been added to show the significant differences between Ca-citrate, Ca-Chloride and Ca-phosphate.

The discussion is correct.

The conclusions are supported by the results shown in this work, and may contribute to clarify the dynamics of foliar absorption by living plants.

Author Response

We do acknowledge the Reviewer for the time spent on a thoughtful revision of our manuscript. Your very meaningful remarks and concerns gave us the opportunity to improve the quality of the text. All your comments were carefully considered, and the response to your point-by-point comments is presented below. 

Point-by-point comments

1- The objective of this work should be more explicitly indicated in the abstract.

Action: We rewrote the 'Abstract' part that mentioned the aims to provide greater clarity. Now it reads: " […] Herein, in vivo microprobe X-ray fluorescence was employed aiming to monitor the foliar absorption of CaCl2, Ca-citrate complex, and Ca3(PO4)2 nanoparticles with and without the use of adjuvant. We also investigated whether Sr2+ can be employed as a Ca2+ proxy in foliar absorption studies […]”

2- Likewise, in the introduction, the objectives of this research have not been clearly stated, in my opinion, so they should be added

Action: We rewrote the paragraph that mentioned the aims to improve its clarity.

We rewrote the 'Abstract' part that mentioned the aims to provide greater clarity. Now it reads:

"Foliar application of Ca is a common practice used by fruit growers to prevent Ca-related disorders and increase fruit quality, however, Ca foliar absorption and transport have not yet been clarified. In this sense, the present study aims to monitor the foliar absorption of distinct Ca sources, that is, salt CaCl2 (Ca- chloride), nanosized Ca3(PO4)2 (Ca- phosphate), and Ca-citrate chelate, by tomato plants. We also investigated the effect of adjuvant of kind mineral oil on the absorption process and the effect on the treatments on the leaf surface"...

3- In the results section in Table 1, no letters , nor * have been added to show the significant differences between Ca-citrate, Ca-Chloride and Ca-phosphate.

Action: No action was taken about this remark since in Table 1 we did not perform statistical analysis because we are not comparing the concentration of the compounds